# SUSTAINABLE RESOURCE MANAGEMENT

**Nicholas Martin**[1], **Ting Sheng Tan**[1], **Peter Hill**[1], and **Francois Buet-Golfouse**[1,2]

[1]Decision Science, JPMorgan Chase, [2]University College London

## ABSTRACT

Given finite resources and growing demand, a supply-side balance must be struck between maximising profit and sustainable resource management. This paper combines the two techniques in a stochastic setting to create a sustainable profit model and uses Gaussian processes to estimate and bound resource dynamics.

## 1 INTRODUCTION

Resource management is an important problem in a finite world. One needs to handle resource exploitation and regeneration simultaneously to ensure that the resource does not run the risk of extinction. We solve this problem within the framework from Dasgupta & Heal (1980), de Vries (2012) and Wackernagel et al. (2021). This is done by modelling the resource size using Gaussian Processes, "G.Ps" given varying sustainability priorities, whilst being able to consider model uncertainty. The developments in this paper particularly apply to the management of forests, fisheries and water supplies, but are not limited to those.

## 2 DYNAMICS OF RENEWABLE RESOURCES

**Classical Worldview** The resource size $R$ is given by the ordinary differential equation ("O.D.E")

$$\frac{dR}{dt} = F(R, G) - H(R, G), \tag{1}$$

where $F$ is the degradation/regeneration function, $G$ the environment and $H$ the *harvest* function. Given a carrying capacity $K$ and regeneration rate $\alpha$, we use the logistic growth model (Allen & McGlade, 1987) $F(R, G) = \alpha R \left(1 - \frac{R}{K}\right)$. This paper uses a Cobb-Douglas function (Mas-Colell et al., 1995), where $H$ depends on the depletion effort, $E$, (Dasgupta & Heal, 1980; Brede & De Vries, 2010)

$$H(R, E) = \varepsilon E^a R^b, \tag{2}$$

with $\varepsilon > 0$ a measure of *effectiveness* and $a, b \in (0, 1)$ are the respective output elasticities of effort and resource size. We use two methods to calculate $E$: profit maximisation and sustainable management.

**Strategic Worldview** Suppose that exploiting companies seek to maximise their profit $\Pi = pH(R, E) - cE = p\varepsilon_\Pi E^a R^b - cE$, where $p > 0$ is the price of a resource's unit and $c > 0$ is the cost of an effort unit. The optimal effort to *maximise profit* can be deduced to be worth

$$E_\Pi = \left(\frac{c}{ap\varepsilon_\Pi}\right)^{-\frac{1}{1-a}} R^{\frac{b}{1-a}}. \tag{3}$$

For *sustainable management*, one might want to ensure that the resources size does not change, i.e., $dR/dt = 0$. In the Cobb-Douglas setting, this is equivalent to

$$E_S = \left(\frac{\alpha}{\varepsilon_S K}\right)^{\frac{1}{a}} R^{\frac{1-b}{a}} (K - R)^{\frac{1}{a}}. \tag{4}$$

One can take a multi-agent approach such that both profit maximisation and sustainability are considered. One can take a weighted, linear combination of the harvest functions accordingly, *sustainable profit*:

$$H(R, E) = (\gamma \epsilon_\Pi E_\Pi^a + (1 - \gamma) \epsilon_S E_S^a) R^b, \tag{5}$$

where $\gamma \in [0, 1]$ weighs profit maximisation and sustainable management.

**Stochastic Worldview** Deterministic systems are limited by: a random resource environment, an unknown form of the degregation/regeneration function and random agent behaviour. Only the first point has received significant attention (May, 1974; Beddington & May, 1977). To tackle this we apply Gaussian Process ("G.Ps") Rasmussen & Williams (2005) to resource modelling. Randomness is introduced in the Euler scheme given that we only observe $R$ at discrete time steps $t_1, \cdots t_n$. The *difference equations* can be written as

$$
\begin{aligned}
R_{t_{k+1}} = R_{t_k} &+ e^{\mu_{t_k}} R_{t_k} \left(1 - R_t e^{-\nu_{t_k}}\right) (t_{k+1} - t_k) \\
&- (\gamma \varepsilon_\Pi E_\Pi^a + (1 - \gamma) \varepsilon_S E_S^a) R_{t_k}^b (t_{k+1} - t_k) + (\omega_{t_{k+1}} - \omega_{t_k}).
\end{aligned}
\tag{6}
$$

Given that we have access to $n'$ observations of $R$, we can infer $R$'s drift numerically. We posit the drift is a random function of time and resource size $\text{drift}_t = H(R_t, t)$. The *perceived* law of motion is therefore $dR_t = H(R_t, t)dt + d\omega_t$, where $\omega$ is a Brownian motion with volatility $\sigma > 0$. Since the drift vector $\mathbf{drift} = \left(\text{drift}_{t_1}, \cdots, \text{drift}_{t_{n'}}\right)$, and a kernel $\mathbf{K}[(t, R_t), (t', R_{t'})]$, we consider a RBF kernel, we obtain a *prior* distribution on the drift vector $\mathbf{drift} \sim N(0, \mathbf{K})$. The overall estimation problem for the drift reduces to a G.P. regression, and gives a closed form solution for $\mathbb{P}(\mathbf{drift}|\mathbf{R})$ (Sarkka & Solin, 2019; Williams & Rasmussen, 2006).

## 3 EXPERIMENTATION

Defining extinction time $t_{\text{ex}} := \inf\{t_k, k \in (1, 1000] : R_{t_k} \leq 0\}$, we choose different values of $\gamma$, run $100,000$ simulations using Equation 6 and record extinction times for each of the simulations. A simulation does not go extinct if $\nexists t_{\text{ex}}$. Following this, G.Ps were used to model the resource drift within a $95\%$ confidence interval, using GPflow (Matthews et al., 2017). The left plot shows that

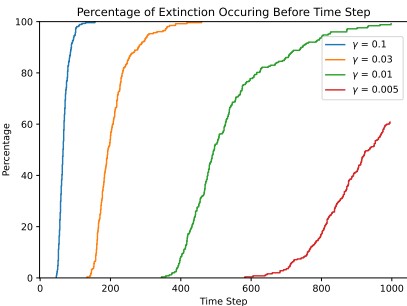 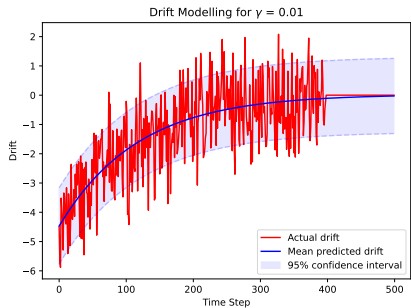

Figure 1: In the left figure, the final value of the plot indicates the proportion of extinction events. The difference between 100 and the final value is the percentage of events that did not go extinct. $\gamma = 0$ was excluded from the plots as no simulations terminated within $1,000$ time steps. In the right figure, we fit a G.P to the resource dynamics and model the mean resource size drift within a confidence bound.

the higher $\gamma$, the greedier the harvest, therefore, the shorter the expected time to extinction. $\gamma$ can be thought of as a measure of one's risk appetite for an extinction event: the higher $\gamma$, the higher the risk appetite and *vice versa*. The right plot bolsters the argument for using G.Ps to model stochastic resource dynamics within a confidence bound, allowing for quantitative resource planning.

## 4 CONCLUSION

In conclusion, this paper introduces a stochastic, sustainable profit model and demonstrates that G.P.s can learn the resource resource dynamics and associated uncertainties. Further research needs to be performed on the convergence of G.Ps on more exotic harvest functions with the aim to use this theory in practice on real world data.

Authors Nicholas Martin, Ting Sheng Tan and Peter Hill meet the URM criteria of ICLR 2023 Tiny Papers Track.

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

APPENDIX

In the following sections, we include further details for our theory and experimentation.

## A    BACKGROUND

The Sustainable Development Goals ("S.D.G.s") put forward by the United Nations [1], albeit only adopted in 2015, have a long history, building on the "Agenda 21" from the Rio Earth Summit in 1992. The 17 goals in the "2030 Agenda for Sustainable Development" include, for instance, Goal#8 centred on decent work and economic growth and Goal#13 on climate action. As illustrated by discussions around planetary boundaries Steffen et al. (2015) or resource scarcity Wackernagel et al. (2021), these two objectives may lead to some necessary trade-offs de Vries (2012).

## B    ANOTHER HARVEST FUNCTION

An example of harvest rate is a pro-rated allocation of the resource per person. In this case $G = P$ where $P$ denotes the population:

$$\frac{dP}{dt} = (b - m)P, \tag{7}$$

where $b$ is the birth and $m$ the mortality rate. This is a model of exponential growth (trivially, $P_t = P_0 e^{(b-m)t}$. One can then choose $H(R, G) = H(R, P) = \beta P$, leading to the (coupled) two-dimensional O.D.E. system:

$$\begin{cases} \frac{dP}{dt} = (b - m)P \\ \frac{dR}{dt} = \alpha R \left(1 - \frac{R}{K}\right) - \beta P. \end{cases} \tag{8}$$

Note that, in general, one does not assume that the resource can become negative, hence $H(R, P) = \beta P \mathbf{1}_{\{R>0\}}$, so that the harvest function is zero if the resource is extinct.

## C    ANOTHER PROFIT MAXIMISATION

**Zero-profit condition**    If we assume competition amongst firms, then either $E_{0\Pi}^* = 0$ or

$$E_{0\Pi}^* = \left(\frac{c}{p\varepsilon}\right)^{-\frac{1}{1-a}} R^{\frac{b}{1-a}} \tag{9}$$

Let us notice that, thanks to the particular structure of the Cobb-Douglas production function, $E_{0\Pi}^* \propto E_{\Pi}^*$.

## D    THEORY

**Profit Maximisation**    Suppose that exploiting companies (e.g., fisheries) seek to maximise their profit

$$\Pi = pH(R, E) - cE = p\varepsilon_{\Pi} E^a R^b - cE, \tag{10}$$

where $p > 0$ is the price of a resource's unit and $c > 0$ is the cost of an effort unit (e.g., labour and capital costs). Thanks to the first-order condition, the optimal effort can be deduced to be worth

$$\frac{d\Pi}{dE} = 0$$
$$\Rightarrow E_{\Pi} = \left(\frac{c}{ap\varepsilon_{\Pi}}\right)^{-\frac{1}{1-a}} R^{\frac{b}{1-a}}. \tag{11}$$

---

[1]cf. https://sdgs.un.org/goals

**Sustainable Management**    A key principle for the renewable resource management is to ensure that its population is constant, i.e., choose the harvest function such that $dR/dt = 0$. In other words, given Equation 1, this implies $H(R, G) = F(R, G)$. In the case of pro-rated allocation, this leads to

$$\beta = \frac{\alpha R(K - R)}{KP}.$$  (12)

Similarly, in the Cobb-Douglas setting, this is equivalent to

$$E_s = \left(\frac{\alpha}{\varepsilon_s K}\right)^{\frac{1}{a}} R^{\frac{1-b}{a}} \left(K - R\right)^{\frac{1}{a}}.$$  (13)

It is immediate to observe that the latter is rather different from the effort derived via profit maximisation $E_\Pi$. Furthermore, It is straightforward to check that $dR/dt < 0$, for $R > 0$, if and only if $E > E_s$. In other words, as expected, any effort above and beyond the sustainable level leads to a depletion of the resource size.

**Gaussian Processes**    An introduction to G.P.s is given in Rasmussen & Williams (2005) and in Chapter 12 of Sarkka & Solin (2019). In short, a G.P. $\mathbf{x}(\boldsymbol{\xi})$ is a random function with $d$-dimensional input $\boldsymbol{\xi}$ such tha any finite collection of random variables $\mathbf{x}(\boldsymbol{\xi}_1), \cdots, \mathbf{x}(\boldsymbol{\xi}_n)$ has a multi-dimensional Gaussian distribution. A G.P. can be defined in terms of a mean $\mathbf{m}(\boldsymbol{\xi})$ and a covariance function (*kernel*) $\mathbf{C}(\boldsymbol{\xi}, \boldsymbol{\xi}')$, which implies that the joint distribution of an arbitrary finite collection of random variables $\mathbf{x}(\boldsymbol{\xi}_1), \cdots, \mathbf{x}(\boldsymbol{\xi}_n)$ follows a multi-dimensional Gaussian distribution:

$$\begin{pmatrix} \mathbf{x}(\boldsymbol{\xi}_1) \\ \vdots \\ \mathbf{x}(\boldsymbol{\xi}_n) \end{pmatrix} \sim N \left( \begin{pmatrix} \mathbf{m}(\boldsymbol{\xi}_1) \\ \vdots \\ \mathbf{m}(\boldsymbol{\xi}_n) \end{pmatrix}, \begin{pmatrix} \mathbf{C}(\boldsymbol{\xi}_1, \boldsymbol{\xi}_1) & \cdots & \mathbf{C}(\boldsymbol{\xi}_1, \boldsymbol{\xi}_n) \\ \vdots & \ddots & \vdots \\ \mathbf{C}(\boldsymbol{\xi}_n, \boldsymbol{\xi}_1) & \cdots & \mathbf{C}(\boldsymbol{\xi}_n, \boldsymbol{\xi}_n) \end{pmatrix} \right).$$  (14)

This is all that is required for our application as data are always finite and computations performed on finite index sets.

Importantly, G.P.s are *non-parametric* tools, thus also allowing more flexible models. Indeed, as pointed out in de Vries (2012) (Chapter 12's Appendix), while the logistic growth model,

$$F(R, G) = \alpha R \left(1 - \frac{R}{K}\right),$$  (15)

it may be qualitatively correct and quantitatively straightforward to manipulate, actual dynamics can be more complicated.

**Introducing Stochasticity**    We introduce randomness in the dynamics of $R$ as follows

$$dR_t = \alpha_t R_t \left(1 - R_t/K_t\right) dt - \left(\gamma \varepsilon_\Pi E_\Pi^a + (1 - \gamma)\varepsilon_s E_s^a\right) R_t^b dt + d\omega_t,$$  (16)

where $\alpha$, $K$ and $\varepsilon$ are now stochastic and $\omega$ represents noise. In addition, we introduce $\mu_t = \log \alpha_t$, $\nu_t = \log K_t$ and model both as G.P.s. To recapitulate, we have included *environmental randomness* via the noise term $\omega$ and *uncertainty* via the G.P.s in the regeneration and exploitation functions. This corresponds to the fact that we do not observe those perfectly and can only estimate them.

**Gaussian Process Estimation**    Lastly, when estimating $R$'s drift, a discretisation of the *perceived* law of motion leads to

$$R_{t_{k+1}} | R_{t_k}, \text{drift}_{t_k} \sim N\left(R_{t_k} + \text{drift}_{t_k}\left(t_{k+1} - t_k\right), \sigma^2\left(t_{k+1} - t_k\right)\right).$$  (17)

## E    EXPERIMENTATION

To understand the overall resource system (corresponding to Eq. 16), we use Monte Carlo simulations to observe its behaviour. We thus produce a toy example with the following parameters for our experiments. For different values of $\gamma$, we run $100,000$ simulations using Equation 6

The regeneration function is chosen according to these settings:

- Initial resource level: $R_0 = 500$,
- Median carrying capacity: $\overline{K} = 1000$,
- Median regeneration rate: $\overline{\alpha} = 0.1$.

On the other hand, the harvest function is specified as follows:

- Cobb-Douglas parameters: $a = b = 0.5$ and $\varepsilon_\Pi = \varepsilon_S = 1$.
- Price of a resource's unit: $p = 2$.
- Price of an effort unit: $c = 1$.

The G.P. specifications are:

- $n = 1000$ equally distributed time-steps with $t_{k+1} - t_k = 1$.
- Covariance matrices are of O.U. type $\mathbf{C}(t, t') = 0.1e^{-|t-t'|}$.
- The mean of the log carrying capacity, $\nu$ is simply $\log \overline{K}$ and the mean of the log regeneration rate, $\mu$ is similarly given by $\log \overline{\alpha}$.
- $\omega$ is a standard Brownian motion (i.e., its volatility is $\sigma = 1$), so that $\omega_{t_{k+1}} - \omega_{t_k} \sim N(0, t_{k+1} - t_k)$.
- Random seed is set to 20230228.

Finally, when modelling the drift the drift for $\gamma = 0.01$:

- We assume the first simulation with random seed 20230228 is our actual drift.
- The trained RBF kernel has a variance of 171.855 and a length scale of 3011.19.

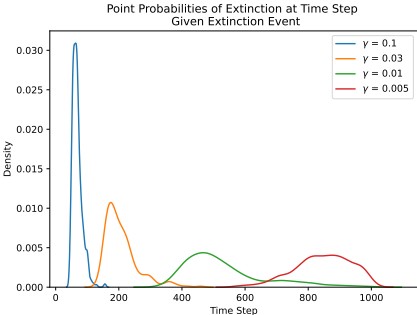

Figure 2: Point probabilities for time of extinction given an extinction event. High gamma induces a greedier harvest function and, thus, the point probabilities of extinction are higher for lower time steps.

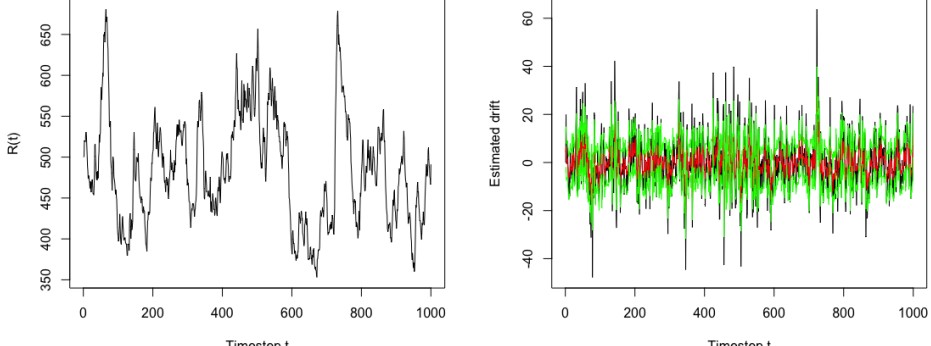

Figure 3: Left: Dynamics of the resource $R$ over 1000 timesteps. Right: True drift (black) of the resource's dynamics, its estimation (red) and the estimation plus or minus one standard deviation (green).

