# OpenReview forum: "Sustainable Resource Management"
_ICLR.cc/2023/TinyPapers — Submitted to Tiny Papers @ ICLR 2023_

### Official Review · Reviewer_BhYP · 2023-03-27

**Confidence:** 3

**Summary Of Contributions:**

In this authors proposed use of stochastic model to learn sustainable resource dynamics with an aim of maximising profit while managing sustainable resources

**Rating:**

High Potential (HP): a submission which meets the reviewing criteria and has potential to make an impact on the field

**Strengths And Weaknesses:**

Strengths:

- Authors targeted an important concern which is the need for time

- The proposed solution is evaluated with a simulation

Weakness:

-Authors did not test the solution with real-world data, or neither the data collected using a standard simulator

**Suggested Changes:**

We suggest authors test the solution with real-world data, or with the data collected using a standard simulator

---

### Author Response · Authors · 2023-06-01
**Opt-in for archival**

The authors wish to opt-in for archival in the ICLR Tiny Papers track.

---

### Meta-Review · Area_Chair_XiUS · 2023-04-07

**Recommendation:** Invite to present
**Confidence:** 4

**Metareview:**

This paper proposes a stochastic model to manage sustainable resources with the goal of maximizing profit. The authors' contributions address an important concern. However, the proposed solution is evaluated through simulations and not tested on real-world data. The strengths of this paper lie in combining two techniques in a stochastic setting.

**Summary:**

The paper presents a novel approach for sustainable resource management in a stochastic model to optimize profit, using two existing techniques. The proposed solution is evaluated with a simulation, but the lack of testing with real-world data or a standard simulator is a weakness that should be addressed.

**Comments And Feedback To The Authors:**

We suggest the authors consider testing the solution with real-world data or with data collected using a standard simulator.

**Reason For Not Giving A Higher Recommendation:**

The paper lacks real-world data testing, which is a weakness that should be addressed. Therefore, we suggest the authors consider testing the solution with real-world data or with data collected using a standard simulator.

**Reason For Not Giving A Lower Recommendation:**

The work explores the crucial problem of maximizing profits while using sustainable resources and practicing sustainable resource management.

---

### Decision · Program_Chairs · 2023-04-09

Invite to present